# Corner Convergence Effect of Enclosed Blast Shock Wave and High-Pressure Range

Xudong Li, Haojie Chen, Jianping Yin * and Zhijun Wang

College of Electromechanical Engineering, North University of China, Taiyuan 030051, China
* Correspondence: yjp123@nuc.edu.cn; Tel.: +86-139-9420-8931

**Abstract:** An explosion inside a cabin will converge at the corners to form high-pressure areas, significantly impacting the destruction of a bulkhead structure. This paper investigates shock wave convergence characteristics at the corners when the explosive detonates at the center of the cabin, based on a combination of the wall reflection law for shock waves and a numerical simulation method. The parameter K represents the aspect ratio of the cabin structure. This study shows that when $1 \leq K \leq 1.19$, the high pressure at the corner is caused by the superposition of Mach waves along both wall surfaces. However, for the initial shock wave, when $1.2 < K \leq 2$, the high pressure is caused by the superposition of Mach waves along the longer wall surface and regular reflected waves on the shorter wall surface; when $2 < K$, the cause are Mach waves along the longer wall surface and the corresponding positive reflection on the shorter wall surface. The influence of K on the range for the high-pressure region at the corner is also analyzed, the functional relationship between the range of the high-pressure area and K is given, and the universality is verified.

**Keywords:** internal explosion; shock wave; corner; structural dimensions; Mach waves





## 1. Introduction

Internal explosions can cause more significant destruction to structures than air explosions due to the combined effect of the reflection, superposition, and convergence of shock waves [1]. High-pressure shock waves can cause structural damage [2]. The pressure peak resulting from the superposition and convergence effect of a shock wave at the corner of a cabin during implosion is significantly higher than the reflected shock wave at the same distance from the wall in an open environment, which can cause a local tear in the corner of the cabin structure first and then expands to the destruction of the entire bulkhead. Therefore, studying the convergence effect of the shock wave at the corner of the cabin during implosion is critical. It is necessary to understand the high-pressure formation rules and the factors influencing the convergence effect at the corner of the implosion shock wave to guide the design of the protection of the cabin structure against internal explosion, and it is also of importance for shock-wave experiments to determine the Hugoniot and melting curves of metals [3,4].

Explosions inside chambers have been a hot topic of research [5–9]. There are a few specific reports on internal blast wave loading [10]. Shock waves have a significant convergence effect at the corners under internal blast conditions [11]. A combined experimental and numerical simulation study [12,13] of the characteristics and typical destruction modes of cabin structures under implosion loads showed that the intensity of the converging shock waves at the corner of two-wall and three-wall surfaces was, respectively, 5- and 12-times greater than the reflected shock waves on the same region of the wall and that the primary failure mechanism of the bulkhead structure during the implosion of the cabin was tearing failure along the corner. Another numerical simulation study of the load situation under implosion conditions showed that the peak pressure at the corner of the three-wall surface was 9–12 times greater than the pressure at the center of the bulkhead, and the

peak pressure at the corner of the two-wall surface was 3–5 times greater than the pressure at the center of the bulkhead [14]. An experimental study measuring the pressure at the corner of a two-wall surface and the peak pressure at the center of the bulkhead showed that the peak pressure at the two walls was smaller than the pressure at the center of the bulkhead [15]. Another study measuring the peak value of the shock wave for three-corner structures (flat plate, concave plate, and convex plate transition connections) at different doses and the peak value of the initial shock wave showed that the corner structure could retard the convergence effect of shock waves at a low dosage [16]; however, when the dosage was higher, the corner structure did not significantly retard the convergence effect. The maximum ratio of the corner converging shock waves to the initial shock waves was 1.24. In addition, another study was reported using an imaging method to explain the convergence effect of corner shock waves [17], whose angle of incidence was the same as the angle of reflection. The actual reflection of the shock wave in the cabin includes the regular oblique reflection and Mach reflection, which was also considered significant. The analysis confirmed the convergence effect of the shock waves at the corner. However, compared to the studies mentioned above, there was a difference between the peak pressures of the convergence of the shock wave at the corner. The study further suggests that the corner convergence occurs at a specific corner area and that the difference in the results is owing to differences in experimental and simulation measurement points. Therefore, it is necessary to investigate the problem of defining the corner convergence area. In addition, the formation of high-pressure areas at the corners and the associated factors have not yet been determined and must be studied in detail.

The characteristics of the explosion load in the enclosed space depend mainly on the spatial dimensions of the structure [18,19]. This paper determines the causes of the convergence phenomenon of shock waves at the corner by 2D cross-sectional analysis with aspect ratio variation. Furthermore, the peak pressure contour map of the corner area at different aspect ratios was plotted through extensive simulation calculations to make a preliminary determination of the range of the high-pressure area at the corner and obtain the functional relationship between the high-pressure area and the size of the structure.

The research in this paper is based on the following three points:
(1) The explosives are in the center of the cabin;
(2) The structure is assumed to be a rigid wall;
(3) The focus is on the peak of the shock wave only.

## 2. Simulation Model

### 2.1. Model Design

When the explosive detonates in the center of the cabin, an arbitrary surface is chosen through the location of the explosion point to intersect, giving the 2D rectangular cross-sectional diagram shown in Figure 1. The dual study of the spread of shock waves on the cross-section and the convergence effect at the corners simplifies the calculation and has a general character. Another study [20] adopted the same method in determining the influential factors for implosion loads.

Figure 2 shows the 2D schematic diagram used in the simulation model. Set wall A as the long side and wall B as the short side. The convergence phenomenon at the corners is studied on half of the cross-section, where a and b are halves of the long and short sides, respectively. The red line area in Figure 2 is the corner area, a square with side length b, and the angle between the shock front and the wall surface is $\Phi$. Each side of the corner area is equally divided into 10 parts, and pressure measurement points are set at the intersections, giving a total of 121 side points arranged as shown in Figure 3.

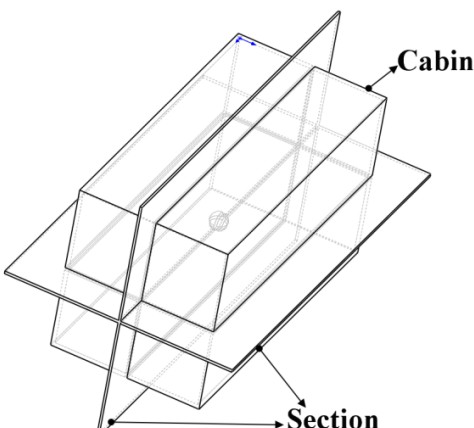

**Figure 1.** Cross-section through the center of the cabin.

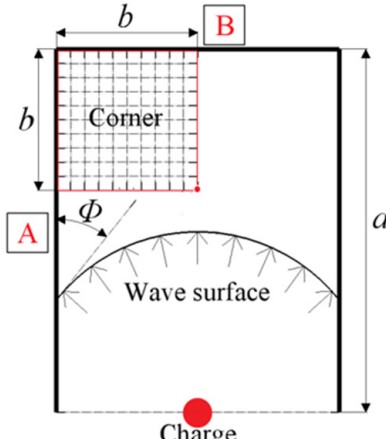

**Figure 2.** Two-dimensional schematic diagram of the calculation model.

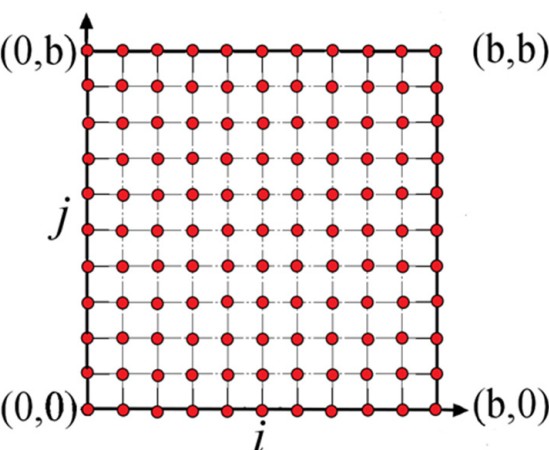

**Figure 3.** Coordinate distribution of measurement points.

The simulation model was run using AUTODYN-2D. The eulerian unit was used for air and the explosives were packed into the air unit. The initial rigid boundary conditions in AUTODYN are adopted for the air boundary to establish a 2D symmetrical model, as shown in Figure 4. The finite element model uses 0.5 mm × 0.5 mm mesh. Simulations were carried out using 0.5 mm, 1 mm, 2 mm, and 4 mm meshes for the shock wave of a

100 g charge at 1 m, indicating that the simulation results converged when the mesh size was 0.5 mm, as shown in Figure 5. The ideal gas equation of state is used for air:

$$P = (\gamma - 1)\rho e \tag{1}$$

where $\gamma$, $\rho$, and $e$ are the specific heat capacity, density, and internal energy of the air, respectively, and the values used for the simulation are $\gamma = 1.4$, $\rho = 1.225 \times 10^{-3}$ g/cm$^3$, and $e = 2.068 \times 10^5$ J.

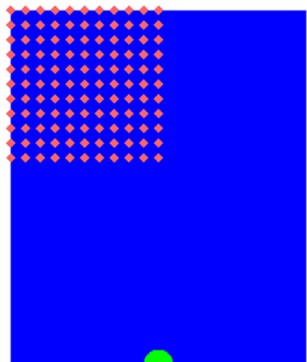

**Figure 4.** Finite element simulation model diagram.

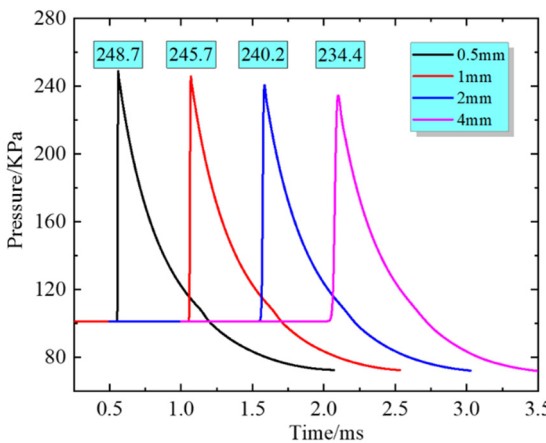

**Figure 5.** Grid convergence checking.

The Jones–Wilkins–Lee (JWL) equation of state is used for the explosive:

$$P_T = C_1\left(1 - \frac{\omega}{r_1 v}\right)e^{-r_1 v} + C_2\left(1 - \frac{\omega}{r_2 v}\right)e^{-r_2 v} + \frac{\omega e}{v} \tag{2}$$

where $C_1$, $C_2$, $r_1$, $r_2$, and $\omega$ are constants, $P_T$, $v$, and $e$ are the pressure, relative volume, and initial energy, respectively. The specific parameters of trinitrotoluene (TNT) are shown in Table 1.

**Table 1.** Parameters of TNT in the JWL equation of state.

| Density, $\rho$ (kg/m$^3$) | Detonation Velocity, $D$ (m/s) | C-J Pressure (Pa) | $C_1$ (Pa) |
|---|---|---|---|
| 1630 | 6800 | $2.10 \times 10^{10}$ | $3.74 \times 10^{11}$ |
| $C_2$ | $r_1$ | $r_2$ | $\omega$ |
| $3.75 \times 10^9$ | 4.15 | 0.9 | 0.35 |

### 2.2. Simulation Model Verification

A related study by Isabelle Sochet [21] investigated an explosion in a partially confined space under different boundary conditions using the 0.106 g equivalent of TNT using gas explosives and obtained a time-history curve of shock wave pressure at each measurement point. The experimental arrangement diagram is shown in Figure 6. This paper uses some of these experimental results to verify the simulation model. The simulation determines the time-history curve of the pressure at measurement points A, B, and C when only one, two, and three walls are available. The model parameters and grid size used in the simulation are identical to those used in Section 2.1. A comparison of the simulation results with the experimental results is shown in Figure 7.

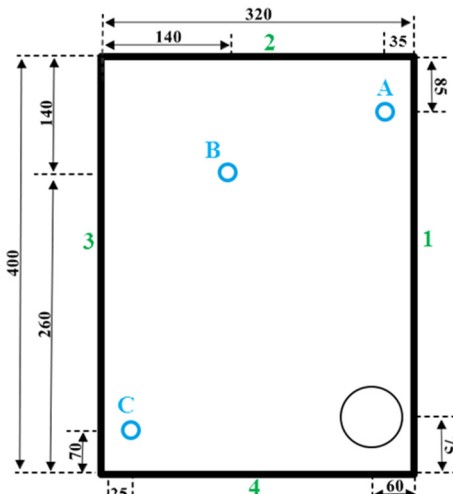

**Figure 6.** Isabelle Sochet experimental layout diagram.

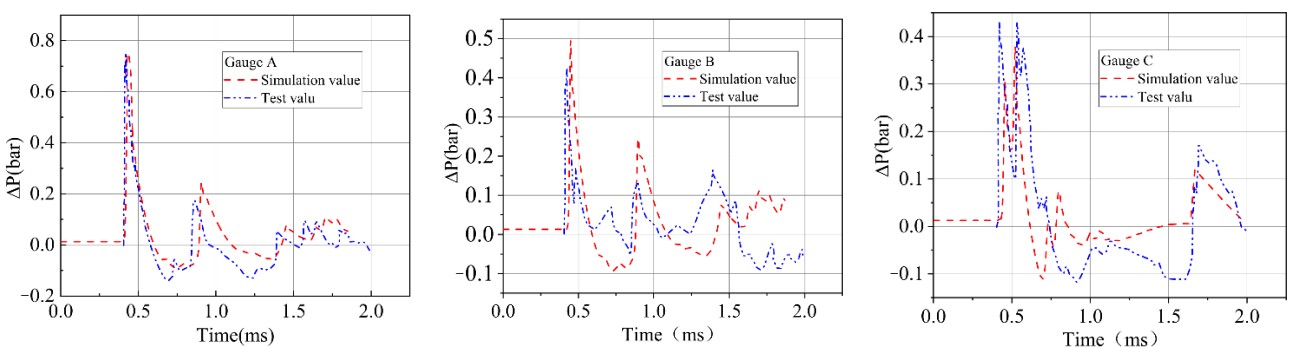

**Figure 7.** Comparison of numerical simulations with experimental results.

From the comparison of the simulation model and experimental results, the pressure peak and the curve change trend are generally consistent, the pressure peak error at B is larger, and the maximum error is 12% which is within the acceptable range and thus, can verify the reliability of the simulation model.

### 2.3. Simulation Working Arrangement

The parameter K represents the ratio of the half of the long side, a, to the half of the short side, b, in Figure 2, viz., K = a/b, which is a dimensionless number used to represent the change in size of the structure. In this study, K is in the range of 1 to 5, and the particular values 500 mm and 1000 mm are used for b. Furthermore, the explosive equivalents of 100 g, 200 g, 500 g, and 1000 g TNT are used. Table 2 gives the specific working conditions.

**Table 2.** Simulation working conditions.

| Serial Number | K | a (mm) | b (mm) | W (g) |
|---|---|---|---|---|
| 1 | 1 | 500 | 500 | |
| 2 | 1.2 | 600 | 500 | |
| 3 | 1.4 | 700 | 500 | |
| 4 | 1.6 | 800 | 500 | |
| 5 | 1.8 | 900 | 500 | |
| 6 | 2.0 | 1000 | 500 | |
| 7 | 2.2 | 1100 | 500 | |
| 8 | 2.4 | 1200 | 500 | 100 |
| 9 | 2.6 | 1300 | 500 | |
| 10 | 2.8 | 1400 | 500 | |
| 11 | 3.0 | 1500 | 500 | |
| 12 | 3.4 | 1700 | 500 | |
| 13 | 4 | 2000 | 500 | |
| 14 | 4.4 | 2200 | 500 | |
| 15 | 5 | 2500 | 500 | |
| 16 | 1.1 | 550 | 500 | |
| 17 | 1.3 | 650 | 500 | |
| 18 | 1.5 | 750 | 500 | |
| 19 | 1.7 | 850 | 500 | |
| 20 | 1.9 | 950 | 500 | 200 |
| 21 | 2.1 | 1050 | 500 | |
| 22 | 2.3 | 1150 | 500 | |
| 23 | 2.5 | 1250 | 500 | |
| 24 | 2.7 | 1350 | 500 | |
| 25 | 2.9 | 1450 | 500 | |
| 26 | 1.1 | 1100 | 1000 | |
| 27 | 1.3 | 1300 | 1000 | |
| 28 | 1.5 | 1500 | 1000 | |
| 29 | 1.7 | 1700 | 1000 | |
| 30 | 1.9 | 1900 | 1000 | 1000 |
| 31 | 2.1 | 2100 | 1000 | |
| 32 | 2.3 | 2300 | 1000 | |
| 33 | 2.5 | 2500 | 1000 | |
| 34 | 2.7 | 2700 | 1000 | |
| 35 | 2.9 | 2900 | 1000 | |

## 3. Mechanism of High-Pressure Formation at Corners

### 3.1. Theoretical Analysis of Convergence Effects at Corners

The spread of an explosive shock wave inside the cabin is complex, characterized by multiple reflections and superpositions, and follows the wall reflection principle. The shock wave reflection at the wall comprises positive and oblique reflections, with the oblique reflections including both regular and Mach reflections [22]. Figure 8 is a schematic diagram of the wall reflection during an air explosion, where d is the vertical distance from the explosive to the wall, c is the distance from the projection point of the explosive on the wall to the intersection of the shock wave front and the wall, $\Phi$ is the angle of incidence of the shock wave on the wall, and $\theta$ is the included angle in the vertical direction between the shock wave front and the wall intersection line. The geometric relationship shows that $\theta = \Phi$, thus $\tan\theta = c/d$, $c = d\tan\theta = d\tan\Phi$, and the Mach angle tends to a limiting value of $39.97°$ [23]. Therefore, when $c/d \geq 0.838$, Mach reflection occurs.

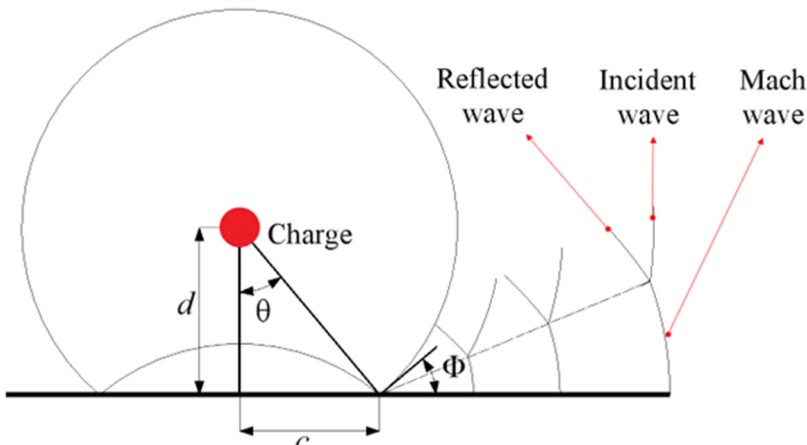

**Figure 8.** Schematic diagram of shock wave wall reflection.

In analogy to Figure 2, the distance from the explosive to bulkhead A is b, and the distance from the explosive to bulkhead B is a. When K = a/b = 1 > 0.838, the reflected shock waves on walls A and B will form Mach reflections before they reach the point (0, b). At point (0, b), the initial shock wave and the Mach reflected waves from walls A and B converge, forming a converging wave. When K > 1, the reflected waves from wall A also form Mach waves before they reach the point (0, b). However, for the reflected waves from wall B, the Mach reflection is formed when b/a ≥ 0.838, i.e., when a/b ≤ 1/0.838 = 1.193. Thus, when 1 ≤ K ≤ 1.193, the initial shock wave at point (0, b) and the Mach reflected waves from walls A and B converge to form a high-pressure region. When 1.193 < K, there is no Mach reflection on wall B. Therefore, the convergence at the point (0, b) is owing to the initial shock wave, the Mach reflection wave from wall A, and the regular reflection wave from wall B. The Mach wavefront gradually widens during its spread, as shown in Figure 8, thus a value exists for n. When K ≥ n, the Mach reflection wave along wall A reaches point (0, b) first, while the initial shock wave superimposes with the Mach wave from wall A in its spread towards point (0, b) and spreads along the three-wave line to wall B without converging at the corner. The simulations in the next section were used to verify the above inference and determine the value of n.

### 3.2. Simulation of Convergence Effects at Corners

Several simulations were conducted with variations in K, as shown in Table 2. As the spread of waves is mainly related to the size of the structure, the convergence clouds of waves at the corner for K = 1, 2, 3, 4, and 5 are exemplified in cases when b = 500 mm and W = 100 g of explosive, as shown in Figure 9.

As observed in the diagram, when K = 1, the converging waves at the corner from the waves reflected at walls A and B and the initial shock waves do not form a noticeable Mach rod phenomenon owing to the short distance between the walls; however, as K increases, the Mach wave on the surface of wall A gradually widens, and a clear Mach rod is observable. As the Mach wave speed is faster than the initial shock wave speed, its wavefront surface gradually flushes with the initial shock wave and surpasses it. Therefore, the high pressure formed near the corner (0, b) comes from the reflection of the Mach wave at the surface of wall B. The above deductions support the theoretical analysis in Section 3.1. For the value of n given in Section 3.1, the simulation shows that when K ≥ 2, the high pressure at the corner (0, b) mainly comes from the positive reflection of Mach waves from wall A to wall B. An example of the spread of the Mach wave and the initial shock wave, when K = 2.4, is shown in Figure 10, where the black dashed line depicts the three-wave line. The high pressure at the corner (0, b) is formed by the positive reflection of the Mach wave.

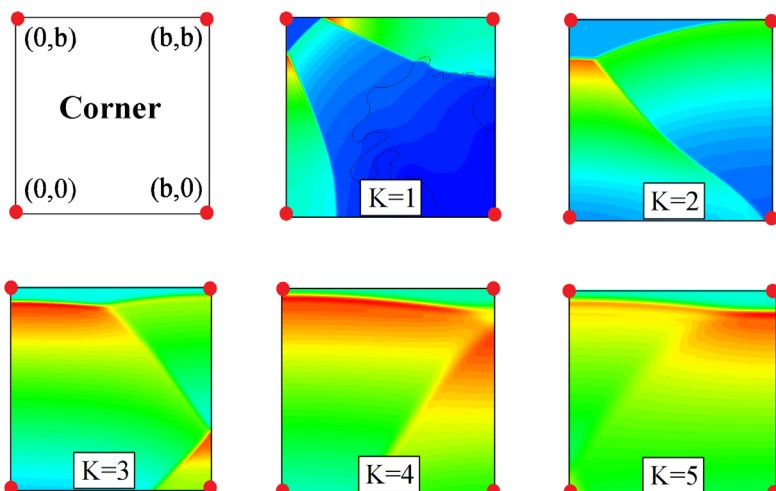

**Figure 9.** The cloud diagram of the convergence effect of shock waves at the corner at different K values when *b* = 500 mm and *W* = 100 g.

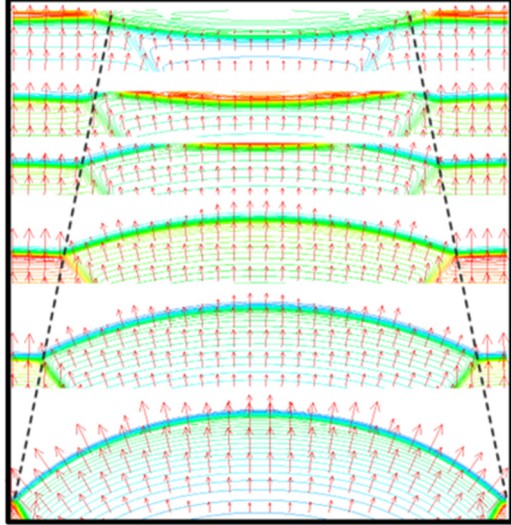

**Figure 10.** The formation process of the convergence effect of shock waves at the corner when K = 2.4.

In previous experiments [2–5], the peak pressure at points (0, b) and (b, b) was frequently used for comparison. The relationship between the peak pressure at the two measuring points and K when b = 500 mm and W = 100 g and the peak pressure ratio at (0, b) and (b, b) is shown in Figure 11. Figure 11 shows that the value for P (b, b)/P (0, b) tends to be stable when K is in the range of 1 to 2.5, but as K increases, the value for P (b, b)/P (0, b) drops linearly, indicating that the corner convergence effect is weakening.

To better demonstrate the changes in the corner high-pressure area with K, the peak pressure distribution in the corner at b = 500 mm and W = 100 g when K = 2, K = 3, K = 4, and K = 5 is shown in Figure 12. The graph shows that K significantly influences the corner's high-pressure area. As K increases, the high-pressure area at the corner gradually expands and moves towards the vicinity of the short side center (b, b) and finally disappears.

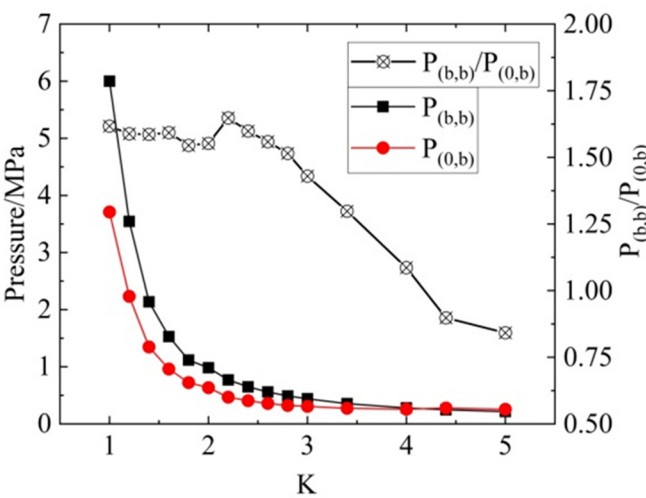

**Figure 11.** Changes in the pressure and pressure ratio at two lateral points with K when b = 500 mm and W = 100 g.

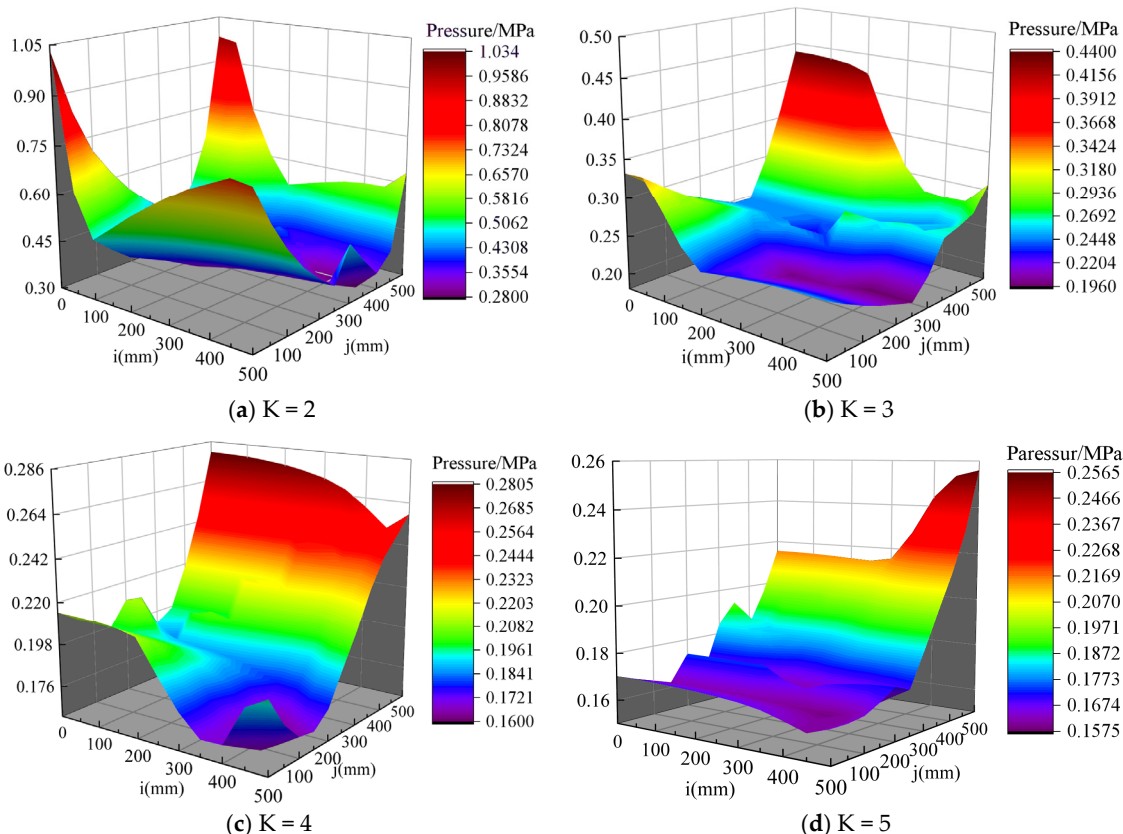

**Figure 12.** Distribution of peak pressure in the corner area at different values of K.

## 4. Determination of the Range of the High-Pressure Area at the Corner

The formation of high pressure at the corner has been studied previously but further study into the range of high-pressure areas is also crucial to understanding the convergence effect of shock waves at the corners. Therefore, the pressure peaks at each measurement point in Figure 3 were recorded and pressure contour maps were plotted. Figure 13 shows the pressure contour map when *b* = 500 mm, *W* = 100 g, and K = 2, 3, 4, and 5.

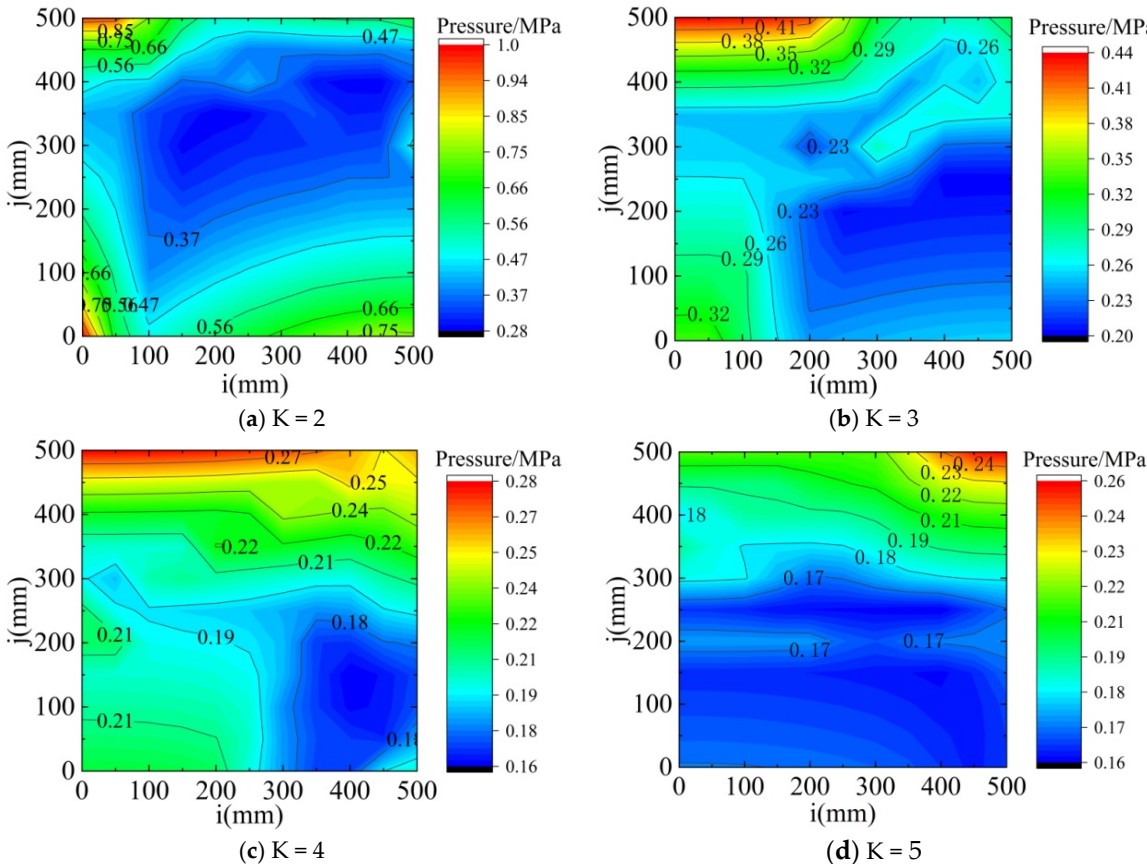

**Figure 13.** Pressure contour distribution diagram in the corner area.

Figure 13 shows that as K increases, the high-pressure area at the corner gradually widens with little change in height. In addition, the high-pressure area gradually moves towards the center of the surface of wall B and is no longer noticeable at the corner but a high-pressure area is formed at the center of the B wall surface. Figure 13 also shows that when K > 3, the high-pressure area at the corner is not apparent; thus, the K range from 1 to 3 will be discussed next. The high-pressure areas at the corners are individually intercepted along the boundary line, as shown in Figure 14, where the numerical relationship between the size of the area boundary and the value of b is indicated.

Figure 14 clearly shows that the high-pressure area appears triangular when K is small and as isosceles triangles when K = 1. As K increases, the shape of the high-pressure area gradually approximates to a rectangular form; this result is consistent with the high-pressure area formation rule at the corner that has already been discussed. The high-pressure area gradually widens primarily as the Mach-reflected wavefront formed at the surface of wall A widens. The relationship between the range of areas obtained in Figure 14 was represented in a coordinate system with the height and width of the corner high-pressure area $a_p$ and $b_p$, respectively. Figure 15 shows the data points in the coordinate system with K, with $a_p/b$ and $b_p/b$ as the horizontal and vertical coordinates, respectively.

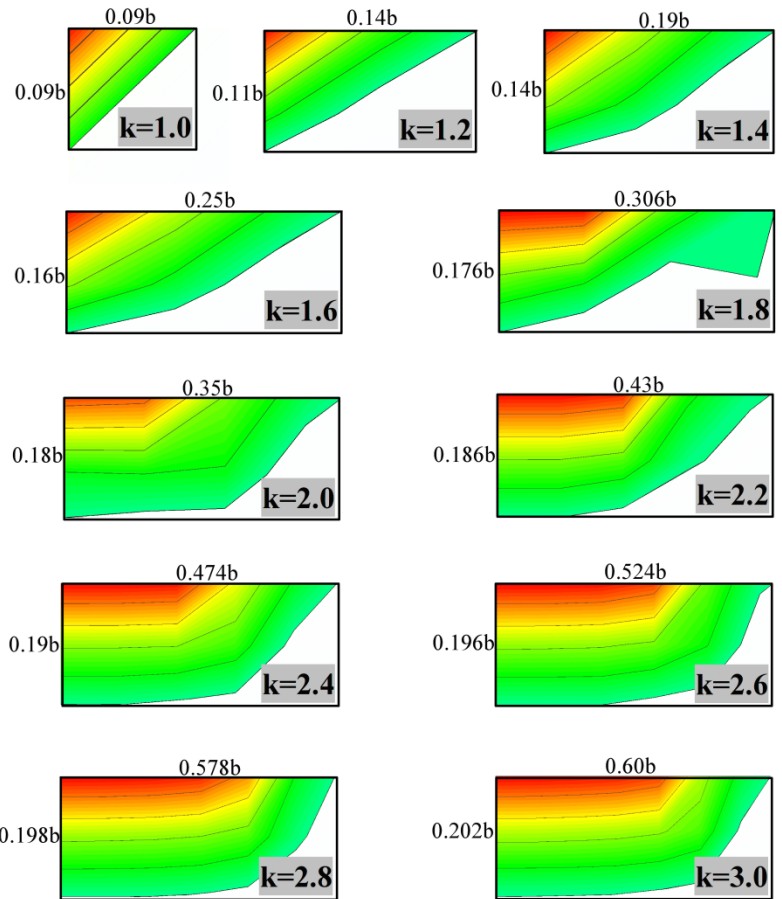

**Figure 14.** Change of the high-pressure area at the corners with the value of K.

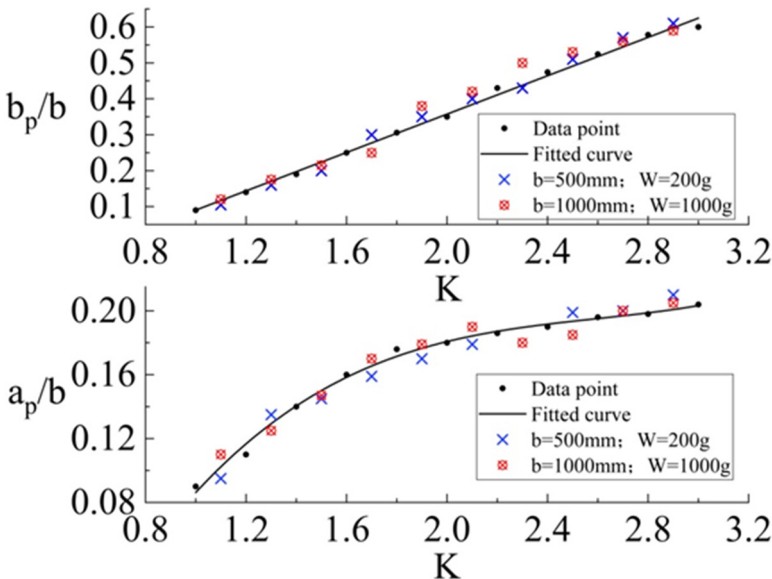

**Figure 15.** Relationship between the size of the high-pressure area and K.

The polynomial fit performed on the data points in Figure 15 shows that $b_p/b$ increases linearly as K increases. Equation (3) gives the functional relationship between $b_p/b$ and K based on the polynomial fit.

$$b_p/b = -0.17673 + 0.26709K \tag{3}$$

The functional relationship between $a_p/b$ and K is also polynomial, as shown by Equation (4):

$$a_p/b = 0.02015K^3 - 0.15693K^2 + 0.42428K - 0.20127 \tag{4}$$

As the fitted data points were obtained under the conditions b = 500 mm and W = 100 g, simulations were conducted considering the generality of the functional relationship given by Equations (3) and (4). Two sets of data points were obtained at b = 500 mm and W = 200 g, and b = 1000 mm and W = 1000 g, as represented in Figure 15. Both data sets satisfy the functional relationship obtained and show that K is the main factor influencing the range of high-pressure areas.

**5. Conclusions**

This study investigated the convergence effect of shock waves at the corners of a cabin under implosion conditions using a 2D cross-sectional method. The high-pressure area formation mechanism during implosion shock wave convergence at the corners and the associated change law were determined with the aspect ratio (K). The specific conclusions are as follows:

1. The aspect ratio, K, significantly influences implosion shock wave convergence at the corner and the associated high-pressure area formation mechanism. When $1 \leq K \leq 1.193$, the convergence of the initial shock wave and Mach reflected waves from the surfaces of walls A and B occurs at the corner, creating a high-pressure region. However, when $1.193 < K < 2$, the convergence at the corner comes from the initial shock wave, the Mach reflection wave on wall A, and the regular reflection wave on wall B. When $2 \leq K$, the high pressure at the corner mainly originates from the positive reflection of Mach waves from the surface of wall A to wall B;

2. As K increases, the convergence effect of the shock waves at the corner is no longer noticeable, and the high-pressure region moves towards the center of the short side;

3. The functional relationship between K and the range of the high-pressure region at the corner was obtained when K = 1 to 3 and its universality was verified.

**Author Contributions:** Conceptualization, X.L. and J.Y.; methodology, X.L.; software, X.L.; validation, X.L.; formal analysis, X.L., J.Y., H.C. and Z.W.; investigation, X.L., J.Y. and Z.W.; data curation, X.L. and J.Y.; writing—original draft preparation, X.L.; writing—review and editing, J.Y.; supervision, J.Y., X.L., Z.W. and H.C.; project administration, X.L. All authors have read and agreed to the published version of the manuscript.

**Funding:** The authors would like to acknowledge the financial support from the 2021 Basic Research Pro-gram of Shanxi Province (Free Exploration), grant number 20210302123207.

**Acknowledgments:** The authors would like to thank the editor, associate editor, and the anonymous reviewers for their helpful comments and suggestions that have improved this paper.

**Conflicts of Interest:** The authors declare no conflict of interest.

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
