# Peer review of "Corner Convergence Effect of Enclosed Blast Shock Wave and High-Pressure Range"

_applsci, doi:10.3390/app122211341_

Round 1

Reviewer 1 Report

This is a very interesting paper where the authors study the shock wave convergence characteristics at the corner when the explosive detonates at the center of a cabin. The reported results are accurate, and the paper is well written. I recommend the publication of the paper after minor changes are made.

Improve the caption of Fig. 1. Now is too laconic. And put it in the same page than the figure.

Put Table 2 in a single page. Do not split it.

Figure 14. I think “ap/b” should be “ap/a”

The same in equation 4,

In the introduction, after the paragraph “It is necessary to understand 29 the high-pressure formation rules and the factors influencing the convergence effect at the 30 corner of the implosion shock wave to guide the design of the protection of the cabin 31 structure against internal explosion.” Please state “It is also of importance for shock-wave experiments used to determine the Hugoniot and melting curves of metals” citing: S. Anzellini et al. Crystals 2021, 11, 452 and S. R. Batty et al. Crystals 2021, 11, 537.

Please check the format of the references and make sure all follow the style of the journal.

Reviewer 2 Report

This paper investigates the convergence of shock waves at the corner of a structure under an enclosed blast and analyses the effect of the aspect ratio of the rectangular closed structure on the convergence mechanism of shock waves. A method for calculating the high-pressure range of the shock wave at the corner after a blast inside a rectangular enclosed structure was developed by fitting simulation data. The results of the study have important engineering applications for the prediction of blast waves inside closed structures, and the paper is well-written. The following changes will be required.

(1) The i and j coordinates in Fig. 11 are not visible, suggesting a change.

(2) It is recommended that a few more citations be given to the relevant literature on the convergence of shock waves at corners.

(3) It is recommended to compare simulation results for different mesh sizes to ensure that the simulations are grid-converged.

(4) It’s suggested that the authors refer to relevant literature to increase the reliability of the results in the paper.

Three-Dimensional Discontinuous Deformation Analysis of Failure Mechanisms and Movement Characteristics of Slope Rockfalls, Rock Mechanics and Rock Engineering; 2022, 55, 275-296.

Experimental investigation on the dynamic mechanical properties and energy absorption mechanism of foam concrete, Construction and Building Materials, 2022, 342:127927.
